# PopArt: Efficient Sparse Regression and Experimental Design for Optimal Sparse Linear Bandits

**Kyoungseok Jang**
University of Arizona
ksajks@arizona.edu

**Chicheng Zhang**
University of Arizona
chichengz@cs.arizona.edu

**Kwang-Sung Jun**
University of Arizona
kjun@cs.arizona.edu

## Abstract

In sparse linear bandits, a learning agent sequentially selects an action and receive reward feedback, and the reward function depends linearly on a few coordinates of the covariates of the actions. This has applications in many real-world sequential decision making problems. In this paper, we propose a simple and computationally efficient sparse linear estimation method called POPART that enjoys a tighter $\ell_1$ recovery guarantee compared to Lasso (Tibshirani, 1996) in many problems. Our bound naturally motivates an experimental design criterion that is convex and thus computationally efficient to solve. Based on our novel estimator and design criterion, we derive sparse linear bandit algorithms that enjoy improved regret upper bounds upon the state of the art (Hao et al., 2020), especially w.r.t. the geometry of the given action set. Finally, we prove a matching lower bound for sparse linear bandits in the data-poor regime, which closes the gap between upper and lower bounds in prior work.

## 1 Introduction

In many modern science and engineering applications, high-dimensional data naturally emerges, where the number of features significantly outnumber the number of samples. In gene microarray analysis for cancer prediction [30], for example, tens of thousands of genes expression data are measured per patient, far exceeding the number of patients. Such practical settings motivate the study of high-dimensional statistics, where certain structures of the data are exploited to make statistical inference possible. One representative example is sparse linear models [19], where we assume that a linear regression task's underlying predictor depends only on a small subset of the input features.

On the other hand, online learning with bandit feedback, due to its practicality in many applications such as online news recommendations [25] or clinical trials [26, 41], has attracted a surge of research interests. Of particular interest is linear bandits, where in $n$ rounds, the learner repeatedly takes an action $A_t$ (e.g., some feature representation of a product or a medicine) from a set of available actions $\mathcal{A} \subset \mathbb{R}^d$ and receives a reward $r_t = \langle \theta^*, A_t \rangle + \eta_t$ as feedback where $\eta_t \in \mathbb{R}$ is an independent zero-mean, $\sigma$-sub-Gaussian noise. Sparsity structure is abundant in linear bandit applications: for example, customers' interests on a product depend only on a number of its key specs; the effectiveness of a medicine only depends on a number of key medicinal properties, which means that the unknown parameter $\theta^*$ sparse; i.e., it has a small number of nonzero entries.

Early studies [2, 8, 24] on sparse linear bandits have revealed that leveraging sparsity assumptions yields bandit algorithms with lower regret than those provided by full-dimensional linear bandit algorithms [3, 4, 11, 1]. However, most existing studies either rely on a particular arm set (e.g., a norm ball), which is unrealistic in many applications, or use computationally intractable algorithms. If we consider an arbitrary arm set, however, the optimal worst-case regret is $\Theta(\sqrt{sdn})$ where $s$ is the sparsity level of $\theta^*$, which means that as long as $n = O(sd)$, there exists an instance for which the algorithm suffers a linear regret [23]. This is in stark contrast to supervised learning

| | REGRET BOUND | DATA-POOR | ASSUMPTIONS |
|---|---|---|---|
| HAO ET AL. [18] | $\tilde{O}(s^{2/3}\mathcal{C}_{\min}^{-2/3}n^{2/3})$ | ✓ | $\mathcal{A}$ SPANS $\mathbb{R}^d$ |
| HAO ET AL. [18] | $\Omega(s^{1/3}\kappa^{-2/3}n^{2/3})$ | ✓ | $\mathcal{A}$ SPANS $\mathbb{R}^d$ |
| ALGORITHM 3 (OURS) | $\tilde{O}(s^{2/3}H_*^{2/3}n^{2/3})$ | ✓ | $\mathcal{A}$ SPANS $\mathbb{R}^d$ |
| THEOREM 5 (OURS) | $\Omega(s^{2/3}\kappa^{-2/3}n^{2/3})$ | ✓ | $\mathcal{A}$ SPANS $\mathbb{R}^d$ |
| HAO ET AL. [18] | $\tilde{O}(\sqrt{\mathcal{C}_{\min}^{-1}sn})$ | ✗ | $\mathcal{A}$ SPANS $\mathbb{R}^d$, MIN. SIGNAL |
| ALGORITHM 4 (OURS) | $\tilde{O}(\sqrt{sn})$ | ✗ | $\mathcal{A}$ SPANS $\mathbb{R}^d$, MIN. SIGNAL |

Table 1: Regret bounds of our work and the prior art where $s$, $d$, $n$ are the sparsity level, the feature dimension, and the number of rounds, respectively. The quantities $\mathcal{C}_{\min}$ and $H_*^2$ are the constants that captures the geometry of the action set (see Eq. (6) and (5)), and $\kappa$ is a parameter for a specific family of arm sets that satisfies $\kappa^{-2} = \Theta(\mathcal{C}_{\min}^{-1}) = \Theta(H_*^2)$. In general, $H_*^2 \leq \mathcal{C}_{\min}^{-1} \leq \mathcal{C}_{\min}^{-2}$ (Propositon 2).

where it is possible to enjoy nontrivial prediction error bounds for $n = o(d)$ [16]. This motivates a natural research question: Can we develop computationally efficient sparse linear bandit algorithms that allow a generic arm set yet enjoy nonvacuous bounds in the data-poor regime by exploiting problem-dependent characteristics?

The seminal work of Hao et al. [18] provides a positive answer to this question. They propose algorithms that enjoy nonvacuous regret bounds with an arbitrary arm set in the data poor regime using Lasso. Specifically, they have obtained a regret bound of $\tilde{O}(\mathcal{C}_{\min}^{-2/3}s^{2/3}n^{2/3})$ where $\mathcal{C}_{\min}$ is an arm-set-dependent quantity. However, their work still left a few open problems. First, their regret upper bound does not match with their lower bound $\Omega(\mathcal{C}_{\min}^{-1/3}s^{1/3}n^{2/3})$. Second, it is not clear if $\mathcal{C}_{\min}$ is the right problem-dependent constant that captures the geometry of the arm set.

In this paper, we make significant progress in high-dimensional linear regression and sparse linear bandits, which resolves or partly answers the aforementioned open problems.

**First** (Section 3), we propose a novel and computationally efficient estimator called POPART (POPulation covariance regression with hARd Thresholding) that enjoys a tighter $\ell_1$ norm recovery bound than the de facto standard sparse linear regression method Lasso in many problems. Motivated by the $\ell_1$ norm recovery bound of POPART, we develop a computationally-tractable design of experiment objective for finding the sampling distribution that minimize the error bound of POPART, which is useful in settings where we have control on the sampling distribution (such as compressed sensing). Our design of experiments results in an $\ell_1$ norm error bound that depends on the measurement set dependent quantity denoted by $H_*^2$ (see Eq. (5) for precise definition) that is provably better than $\mathcal{C}_{\min}^{-1}$ that appears in the $\ell_1$ norm error bound used in Hao et al. [18], thus leading to an improved planning method for sparse linear bandits. **Second** (Section 4), Using POPART, we design new algorithms for the sparse linear bandit problem, and improve the regret upper bound of prior work [18]; see Table 1 for the summary. **Third** (Section 5), We prove a matching lower bound in data-poor regime, showing that the regret rate obtained by our algorithm is optimal. The key insight in our lower bound is a novel application of the algorithmic symmetrization technique [34]. Unlike the conjecture of Hao et al. [18, Remark 4.5], the improvable part was not the algorithm but the lower bound for sparsity $s$.

We empirically verify our theoretical findings in Section 6 where POPART shows a favorable performance over Lasso. Finally, we conclude our paper with future research enabled by POPART in Section 7. For space constraint, we discuss related work in Appendix A but closely related studies are discussed in depth throughout the paper.

## 2 Problem Definition and Preliminaries

**Sparse linear bandits.** We study the sparse linear bandit learning setting, where the learner is given access to an action space $\mathcal{A} \subset \{a \in \mathbb{R}^d : \|a\|_\infty \leq 1\}$, and repeatedly interacts with the environment as follows: at each round $t = 1, \ldots, n$, the learner chooses some action $A_t \in \mathcal{A}$, and receives reward feedback $r_t = \langle \theta^*, A_t \rangle + \eta_t$, where $\theta^* \in \mathbb{R}^d$ is the underlying reward predictor, and $\eta_t$ is an independent zero-mean $\sigma$-subgaussian noise. We assume that $\theta^*$ is $s$-sparse; that is, it has at most $s$ nonzero entries. The goal of the learner is to minimize its pseudo-regret defined as

$$\text{Reg}(n) = n \max_{a \in \mathcal{A}} \langle \theta^*, a \rangle - \sum_{t=1}^n \langle \theta^*, A_t \rangle.$$

**Experimental design for linear regression.** In the experimental design for linear regression problem, one has a pool of unlabeled examples $\mathcal{X}$, and some underlying predictor $\theta^*$ to be learned. Querying the label of $x$, i.e. conducting experiment $x$, reveals a random label $y = \langle \theta^*, x \rangle + \eta$ associated with it, where $\eta$ is a zero mean noise random variable. The goal is to accurately estimate $\theta^*$, while using as few queries $x$ as possible.

**Definition 1.** (Population covariance matrix $Q$) Let $\mathcal{P}(\mathcal{X})$ be the space of probability measures over $\mathcal{X}$ with the Borel $\sigma$-algebra, and define the population covariance matrix for the distribution $\mu \in \mathcal{P}(\mathcal{X})$ as follows:

$$Q(\mu) := \int_{a \in \mathcal{X}} aa^\top d\mu(a) \tag{1}$$

Classical approaches for experimental design focus on finding a distribution $\mu$ such that its induced population covariance matrix $Q(\mu)$ has properties amenable for building a low-error estimator, such as D-, A-, G-optimality [14].

**Compatibility condition for Lasso.** For a positive definite matrix $\Sigma \in \mathbb{R}^{d \times d}$ and a sparsity level $s \in [d] := \{1, \ldots, d\}$, we define its compatibility constant $\phi_0^2(\Sigma, s)$ as follows:

$$\phi_0^2(\Sigma, s) := \min_{S \subseteq [d] : |S| = s} \min_{v : \|v_S\|_1 \le 3 \|v_{-S}\|_1} \frac{s v^\top \Sigma v}{\|v_S\|_1^2}, \tag{2}$$

where $v_S \in \mathbb{R}^d$ denotes the vector that agrees with $v$ in coordinates in $S$ and $0$ everywhere else and $v_{-S} \in \mathbb{R}^d$ denotes $v - v_S$.

**Notation.** Let $e_i$ be the $i$-th canonical basis vector. We define $[x] = \{1, 2, \ldots, x\}$. Let $\mathrm{supp}(\theta)$ be the set of coordinate indices $i$ where $\theta_i \ne 0$. We use $a \lesssim b$ to denote that there exists an absolute constant $c$ such that $a \le cb$.

## 3 Improved Linear Regression and Experimental Design for Sparse Models

In this section, we discuss our novel sparse linear estimator POPART for the setting where the population covariance matrix is known and show its strong theoretical properties. We then present a variation of POPART called WARM-POPART that amends a potential weakness of POPART, followed by our novel experimental design for POPART and discuss its merit over prior art.

**POPART (POPulation covariance regression with hARd Thresholding).** Unlike typical estimators for the statistical learning setup, our main estimator POPART described in Algorithm 1 takes the population covariance matrix denoted by $Q$ as input. We summarize our assumption for POPART.

**Assumption 1.** (Assumptions on the input of POPART) There exists $\mu$ such that the input data points $\{(X_t, Y_t)\}_{t=1}^n$ satisfy that $X_t \overset{\text{i.i.d.}}{\sim} \mu$ and $Q = Q(\mu) := \mathbb{E}_{X \sim \mu}[XX^\top]$. Furthermore, $Y_t = \langle \theta^*, X_t \rangle + \eta_t$ with $\eta_t$ being zero-mean $\sigma$-subgaussian noise. Also, $R_0 \ge \max_{a \in \mathcal{A}} |\langle a, \theta^* - \theta_0 \rangle|$.

---

**Algorithm 1** POPART (POPulation covariance regression with hARd Thresholding)

1: **Input:** Samples $\{(X_t, Y_t)\}_{t=1}^n$, the population covariance matrix $Q \in \mathbb{R}^{d \times d}$, pilot estimator $\theta_0 \in \mathbb{R}^d$, an upper bound $R_0$ of $\max_{a \in \mathcal{A}} |\langle a, \theta^* - \theta_0 \rangle|$, failure rate $\delta$.
2: **Output:** estimator $\hat{\theta}$
3: **for** $t = 1, \ldots, n$ **do**
4:     $\tilde{\theta}_t = Q^{-1} X_t (Y_t - \langle X_t, \theta_0 \rangle) + \theta_0$
5: **end for**
6: $\forall i \in [d], \theta_i' = \mathsf{Catoni}(\{\tilde{\theta}_{ti} := \langle \tilde{\theta}_t, e_i \rangle\}_{t=1}^n, \alpha_i, \frac{\delta}{2d})$ where $\alpha_i := \sqrt{\dfrac{2 \log \frac{2d}{\delta}}{n(R_0^2 + \sigma^2)(Q^{-1})_{ii}(1 + \frac{2 \log \frac{2d}{\delta}}{n - 2 \log \frac{2d}{\delta}})}}$
7: $\hat{\theta} \leftarrow \mathsf{clip}_\lambda(\theta') := [\theta_i' \mathbb{1}(|\theta_i'| > \lambda_i)]_{i=1}^d$ where $\lambda_i$ is defined in Proposition 1.
8: **return** $\hat{\theta}$

---

POPART consists of several stages. In the first stage, for each $(X_t, Y_t)$ pair, we create a one-sample estimator $\tilde{\theta}_t$ (step 4). The estimator, $\tilde{\theta}_t$, can be viewed as a generalization of doubly-robust estimator [10, 12] for linear models. Specifically, it is the sum of two parts: one is the pilot estimator

$\theta_0$ that is a hyperparameter of POPART; the other is $Q(\mu)^{-1}X_t(Y_t - \langle X_t, \theta_0 \rangle)$, an unbiased estimator of the difference $\theta^* - \theta_0$. Thus, it is not hard to see that $\tilde{\theta}_t$ is an unbiased estimator of $\theta^*$. As we will see in Theorem 1, the variance of $\tilde{\theta}_t$ is smaller when $\theta_0$ is closer to $\theta^*$, showing the advantage of allowing a pilot estimator $\theta_0$ as input. If no good pilot estimator is available a priori, one can set $\theta_0 = 0$.

From the discussion above, it is natural to take an average of $\tilde{\theta}_t$. Indeed, when $n$ is large, the population covariance matrix $Q(\pi)$ is close to empirical covariance matrix $\hat{Q} := \frac{1}{n}\sum_{t=1}^{n} X_t X_t^\top$, which makes $\hat{\theta}_{\text{avg}} := \frac{1}{n}\sum_{t=1}^{n}\tilde{\theta}_t$ close to the ordinary least squares estimator $\hat{\theta}_{\text{OLS}} = \hat{Q}^{-1}(\frac{1}{n}\sum_{t=1}^{n} X_t Y_t)$. However, for technical reasons, the concentration property of $\tilde{\theta}_{\text{avg}}$ is hard to establish. This motivates POPART's second stage (step 6), where, for each coordinate $i \in [d]$, we employ Catoni's estimator [27] (see Appendix B for a recap) to obtain an intermediate estimate for each $\theta_i^*$, namely $\theta_i'$.

To use Catoni's estimator, we need to have an upper bound of the variance of $\theta_i'$ for its $\alpha_i$ parameter. A direct calculation yields that, for all $i \in [d]$ and $t \in [n]$,

$$\text{Var}(\tilde{\theta}_{ti}) \leq \left( \max_{a \in \mathcal{A}} \langle \theta^* - \theta_0, a \rangle^2 + \sigma^2 \right) \max_i (Q(\mu)^{-1})_{ii}$$

where $\tilde{\theta}_{ti} := \langle \tilde{\theta}_t, e_i \rangle$. This implies that $(R_0^2 + \sigma^2) \max_i (Q(\mu)^{-1})_{ii}$ is an upper bound of $\text{Var}(\tilde{\theta}_{ti})$. By the standard concentration inequality of Catoni's estimator (see Lemma 1), we obtain the following estimation error guarantee for $\theta_i'$; the proof can be found in Appendix C.1. Hereafter, all proofs are deferred to appendix unless noted otherwise.

**Proposition 1.** Suppose Assumption 1 holds. In POPART, for $i \in [d]$, if $n \geq 2\ln\frac{2d}{\delta}$, the following inequality holds with probability $1 - \frac{\delta}{d}$:

$$|\theta_i' - \theta_i^*| < \sqrt{\frac{4(R_0^2 + \sigma^2)(Q(\mu)^{-1})_{ii}^2}{n} \log \frac{2d}{\delta}} =: \lambda_i$$

Proposition 1 shows that, for each coordinate $i$, $(\theta_i' - \lambda_i, \theta_i' + \lambda_i)$ forms a confidence interval for $\theta_i^*$. Therefore, if $0 \notin (\theta_i' - \lambda_i, \theta_i' + \lambda_i)$, we can infer that $\theta_i^* \neq 0$, i.e., $i \in \text{supp}(\theta^*)$. Based on the observation above, POPART's last stage (step 7) performs a hard-thresholding for each of the coordinates of $\theta'$, using the threshold $\lambda_i$ for coordinate $i$. Thanks to the thresholding step, with high probability, $\hat{\theta}$'s support is contained in that of $\theta^*$, which means that all coordinates $i$ outside the support of $\theta^*$ (typically the vast majority of the coordinates when $s \ll d$) satisfy $\hat{\theta}_i = \theta_i^* = 0$. Meanwhile, for coordinate $i$'s in $\text{supp}(\theta^*)$, the estimated value $\hat{\theta}_i$ is not too far from $\theta_i^*$.

The following theorem states POPART's estimation error bound in terms of its output $\hat{\theta}$'s $\ell_\infty$, $\ell_0$, and $\ell_1$ errors, respectively. We remark that replacing hard thresholding in the last stage with soft thresholding enjoys similar guarantees.

**Theorem 1.** Take Assumption 1. Let $H^2(Q) := \max_{i \in [d]}(Q^{-1})_{ii}$. Then, POPART has the following guarantees with probability at least $1 - \delta$:

(i) $\forall i \in [d], |\hat{\theta}_i - \theta_i^*| < 2\sqrt{\frac{4(R_0^2 + \sigma^2)(Q(\mu)^{-1})_{ii}}{n} \log \frac{2d}{\delta}}$ so $\|\hat{\theta} - \theta^*\|_\infty < 2\sqrt{\frac{4(R_0^2 + \sigma^2)H^2(Q(\mu))}{n} \log \frac{2d}{\delta}}$,

(ii) $\text{supp}(\hat{\theta}) \subset \text{supp}(\theta^*)$ so $\|\hat{\theta} - \theta^*\|_0 \leq s$,

(iii) $\|\hat{\theta} - \theta^*\|_1 \leq 2s\sqrt{\frac{4(R_0^2 + \sigma^2)H^2(Q(\mu))}{n} \log \frac{2d}{\delta}}$

Interestingly, POPART has no false positive for identifying the sparsity pattern and enjoys an $\ell_\infty$ error bound, which is not available from Lasso, to our knowledge. Unfortunately, a direct comparison with Lasso is nontrivial since the largest compatibility constant $\phi_0^2(\hat{\Sigma}, s)$ is defined as the solution of the optimization problem (2), let alone the fact that $\phi_0^2(\hat{\Sigma}, s)$ is a function of the empirical covariance matrix. While we leave further investigation as future work, our experiment results in Section 6 suggest that there might be a case where POPART makes a meaningful improvement over Lasso.

*Proof of Theorem 1.* Let $\lambda := \max_i \lambda_i = \sqrt{\frac{4(R_0^2 + \sigma^2)H^2(Q(\mu))}{n} \log \frac{2d}{\delta}}$ From Proposition 1 and the union bound, one can check that

$$\|\theta' - \theta^*\|_\infty < \lambda \tag{3}$$

with probability $1 - \delta$. Therefore, the coordinates in $\mathrm{supp}(\theta^*)^c$ will be thresholded out because of $\|\theta' - \theta^*\|_\infty \leq \lambda$. Therefore, (ii) holds and for all $i \in \mathrm{supp}(\theta^*)^c$, $|\hat{\theta}_i - \theta_i^*| = 0$.

By definition, $\hat{\theta} = \mathsf{clip}_\lambda(\theta')$, we can say that $\|\hat{\theta} - \theta'\|_\infty \leq \lambda$. Plus, by Eq. (3), $\|\theta' - \theta^*\|_\infty \leq \lambda$. By the triangle inequality, $\|\theta^* - \hat{\theta}\|_\infty \leq 2\lambda$. Therefore, (i) holds.

Lastly, (iii) can be argued as follows:

$$\|\hat{\theta} - \theta^*\|_1 = \sum_{i \in [d]} |\hat{\theta}_i - \theta_i^*| \leq \sum_{i \in \mathrm{supp}(\theta^*)^c} 0 + \sum_{i \in \mathrm{supp}(\theta^*)} 2\lambda \leq 2s\lambda. \qquad \square$$

**WARM-POPART: Improved guarantee by warmup.** One drawback of the POPART estimator is that its estimation error scales with $\sqrt{R_0^2 + \sigma^2}$, which can be very large when $R_0$ is large. One may attempt to use the fact that POPART allows a pilot estimator $\theta_0$ to address this issue since $R_0$ gets smaller as $\theta_0$ is closer to $\theta^*$. However, it is a priori unclear how to obtain a $\theta_0$ close to $\theta^*$ as $\theta^*$ is the unknown parameter that we wanted to estimate in the first place.

To get around this "chicken and egg" problem, we propose to introduce a warmup stage, which we call WARM-POPART (Algorithm 2). WARM-POPART consists of two stages. For the first warmup stage, the algorithm runs POPART with the zero vector as the pilot estimator and with the first half of the samples to obtain a coarse estimator denoted by $\hat{\theta}_0$ which guarantees that for large enough $n_0$, $\|\hat{\theta}_0 - \theta^*\|_1 \leq \sigma$. In the second stage, using $\hat{\theta}_0$ as the pilot estimator, it runs POPART on the remaining half of the samples.

---

**Algorithm 2** WARM-POPART

1: **Input:** Samples $\{(X_t, Y_t)\}_{t=1}^{n_0}$, the population covariance matrix $Q \in \mathbb{R}^{d \times d}$, an upper bound $R_{\max}$ of $\max_{a \in \mathcal{A}} |\langle \theta^*, a \rangle|$, number of samples $n_0$, failure rate $\delta$.
2: **Output:** $\hat{\theta}$, an estimate of $\theta^*$
3: Run POPART$(\{(X_i, Y_i)\}_{i=1}^{\lfloor n_0/2 \rfloor}, Q, \vec{0}, \delta, R_{\max})$ to obtain $\hat{\theta}_0$, a coarse estimate of $\theta^*$ for the next step.
4: Run POPART$(\{(X_i, Y_i)\}_{i=\lfloor n_0/2 \rfloor+1}^{n_0}, Q, \hat{\theta}_0, \delta, \sigma)$ to obtain $\hat{\theta}$, an estimate of $\theta^*$.

---

The following corollary states the estimation error bound of the output estimator $\hat{\theta}$. Compared with POPART's $\ell_1$ recovery guarantee, WARM-POPART's $\ell_1$ recovery guarantee (Equation (4)) has no dependence on $R_{\max}$; its dependence on $R_{\max}$ only appears in the lower bound requirement for $n_0$.

**Corollary 1.** Take Assumption 1 without the condition on $R_0$. Assume that $R_{\max} \geq \max_{a \in \mathcal{A}} |\langle a, \theta^* \rangle|$, and $n_0 > \frac{32s^2(R_{\max}^2 + \sigma^2)H^2(Q(\mu))}{\sigma^2} \log \frac{2d}{\delta}$. Then, WARM-POPART has, with probability at least $1 - 2\delta$,

$$\|\hat{\theta} - \theta^*\|_1 \leq 8s\sigma \sqrt{\frac{H^2(Q(\mu))\ln\frac{2d}{\delta}}{n_0}}. \tag{4}$$

**Remark 1.** In Algorithm 2, we choose POPART as our coarse estimator, but we can freely change the coarse estimation step (step 3) to other principled estimation methods (such as Lasso) without affecting the main estimation error bound (4); the only change will be the lower bound requirement of $n_0$ to another problem-dependent constant.

**Remark 2.** WARM-POPART requires the knowledge of $R_{\max}$, an upper bound of $\max_{a \in \mathcal{A}} |\langle \theta^*, a \rangle|$; this requirement can be relaxed by changing the last argument of the coarse estimation step (step 3) from $R_{\max}$, to some function $f(n_0)$ such that $f(n_0) = \omega(1)$ and $f(n_0) = o(\sqrt{n_0})$ (say, $\sigma n_0^{\frac{1}{4}}$); with this change, a result analogous to Corollary 1 can be proved with a different lower bound requirement of $n_0$.

**A novel and efficient experimental design for sparse linear estimation.** In the experimental design setting where the learner has freedom to design the underlying sampling distribution $\mu$, the $\ell_1$ error bound of POPART and WARM-POPART naturally motivates a design criterion. Specifically, we can choose $\mu$ that minimizes $H^2(Q(\mu))$, which gives the lowest estimation error guarantee. We denote the optimal value of $H^2(Q(\mu))$ by

$$H_*^2 := \min_{\mu \in \mathcal{P}(\mathcal{A})} \max_{i \in [d]} (Q(\mu)^{-1})_{ii}. \tag{5}$$

The minimization of $H^2(Q(\mu))$ is a convex optimization problem, which admits efficient methods for finding the solution. Intuitively, $H_*^2$ captures the geometry of the action set $\mathcal{A}$.

To compare with previous studies that design a sampling distribution for Lasso, we first review the standard $\ell_1$ error bound of Lasso.

**Theorem 2.** (Buhlmann and van de Geer [6, Theorem 6.1]) With probability at least $1 - 2\delta$, the $\ell_1$-estimation error of the optimal Lasso solution $\hat{\theta}_{\text{Lasso}}$ [6, Eq. (2.2)] with $\lambda = \sqrt{2\log(2d/\delta)/n}$ satisfies

$$\|\hat{\theta}_{\text{Lasso}} - \theta^*\|_1 \leq \frac{s\sigma}{\phi_0^2(\hat{\Sigma}, s)}\sqrt{\frac{2\log(2d/\delta)}{n}},$$

where $\phi_0(\hat{\Sigma}, s)^2$ is the compatibility constant with respect to the empirical covariance matrix $\hat{\Sigma} = \frac{1}{n}\sum_{t=1}^n X_t X_t^\top$ and the sparsity $s$ in Eq. (2).

Ideally, for Lasso, experiment design which minimizes the compatibility constant will guarantee the best estimation error bound within a fixed number of samples $n$. However, naively, the computation of the compatibility constant is intractable since Eq. (2) is a combinatorial optimization problem which is usually difficult to compute. One simple approach taken by Hao et al. [18] is to use the following computationally tractable surrogate of $\phi_0^2(\hat{\Sigma}, s)$:

$$\mathcal{C}_{\min} := \max_{\mu \in \mathcal{P}(\mathcal{A})} \lambda_{\min}(Q(\mu)) \tag{6}$$

where $\lambda_{\min}(A)$ denotes the minimum eigenvalue of a matrix $A$. With the choice of sampling distribution $\mu = \operatorname*{argmax}_{\mu \in \mathcal{P}(\mathcal{A})} \lambda_{\min}(Q(\mu))$, and $n \geq \tilde{\Omega}(\frac{s \cdot \text{polylog}(d)}{\mathcal{C}_{\min}^2})$, with high probability, $\phi_0^2(\hat{\Sigma}, s) \geq \mathcal{C}_{\min}/2$ holds [33, Theorem 1.8], and one can replace $\phi_0(\hat{\Sigma}, s)$ to $\mathcal{C}_{\min}/2$ in Theorem 2 to get the following corollary:

**Corollary 2.** With probability at least $1 - \exp(-cn) - 2\delta$ for some universal constant $c$, the $\ell_1$-estimation error of the optimal Lasso solution $\hat{\theta}_{\text{Lasso}}$ satisfies

$$\|\hat{\theta}_{\text{Lasso}} - \theta^*\|_1 \leq \frac{2s\sigma}{\mathcal{C}_{\min}}\sqrt{\frac{2\log(2d/\delta)}{n}}, \tag{7}$$

The following proposition shows that our estimator has a better error bound compared to the surrogate experimental design for Lasso of Hao et al. [18].

**Proposition 2.** We have $H_*^2 \leq \mathcal{C}_{\min}^{-1} \leq dH_*^2$. Furthermore, there exist arm sets for which either of the inequalities is tight up to a constant factor.

Therefore, our new estimator has $\ell_1$ error guarantees at least a factor $\mathcal{C}_{\min}^{-1/2}$ better than that provided by [18], as follows: when we choose the $\mu$ as the solution of the Eq. (5), then

$$(\text{RHS of (4)}) \lesssim s\sigma H_* \sqrt{\frac{\ln(2d/\delta)}{n}} \lesssim s\sigma \mathcal{C}_{\min}^{-1/2}\sqrt{\frac{\ln(2d/\delta)}{n}} \lesssim s\sigma \mathcal{C}_{\min}^{-1}\sqrt{\frac{\ln(2d/\delta)}{n}} \lesssim (\text{RHS of (7)})$$

In addition, we also prove that there exists a case where our estimator has an $d/s$-order better error bound compared to the traditional lasso bound in Theorem 2, although this is not in terms of the compatibility constant of the empirical covariance matrix $\hat{\Sigma}$.

**Proposition 3.** There exists an action set $\mathcal{A}$ and an absolute constant $C_1 > 0$ such that

$$H_* < C_1 \frac{s}{d} \times \frac{1}{\phi_0^2(\Sigma, s)}$$

For the detailed proof about Proposition 2 and Proposition 3, see Section D in Appendix.

# 4 Improved Sparse Linear Bandits using WARM-POPART

We now apply our new WARM-POPART sparse estimation algorithm to design new sparse linear bandit algorithms. Following prior work [18], we adopt the classical Explore-then-Commit (ETC) framework for algorithm design, and use POPART with experimental design to perform exploration. As we will see, the tighter $\ell_1$ estimation error bound of our POPART-based estimators helps us obtain an improved regret bound.

---

**Algorithm 3** Explore then commit with WARM-POPART

1: Input: time horizon $n$, action set $\mathcal{A}$, warm-up exploration length $n_0$, failure rate $\delta$, reward threshold parameter $R_{\max}$, an upper bound of $\max_{a \in \mathcal{A}} |\langle \theta^*, a \rangle|$.
2: Solve the optimization problem in Eq. (5) and denote the solution as $\mu_*$
3: **for** $t = 1, \dots, n_0$ **do**
4:     Independently pull the arm $A_t$ according to $\mu_*$ and receives the reward $r_t$
5: **end for**
6: Run WARM-POPART($\{A_t\}_{t=1}^{n_0}, \{r_t\}_{t=1}^{n_0}, Q(\mu_*), \delta, R_{\max}$) to obtain $\hat{\theta}$, an estimate of $\theta^*$.
7: **for** $t = n_0 + 1, \dots, n$ **do**
8:     Take action $A_t = \arg\max_{a \in \mathcal{A}} \langle \hat{\theta}, a \rangle$, receive reward $r_t = \langle \theta^*, A_t \rangle + \eta_t$
9: **end for**

---

**Sparse linear bandit with WARM-POPART.** Our first new algorithm, Explore then Commit with WARM-POPART (Algorithm 3), proceeds as follows. For the exploration stage, which consists of the first $n_0$ rounds, it solves the optimization problem (5) to find $\mu_*$, the optimal sampling distribution for POPART and samples from it to collect a dataset for the estimation of $\theta^*$. Then, we use this dataset to compute the WARM-POPART estimator $\hat{\theta}$. Finally, in the commit stage, which consists of the remaining $n - n_0$ rounds, we take the greedy action with respect to $\hat{\theta}$. We prove the following regret guarantee of Algorithm 3:

**Theorem 3.** If Algorithm 3 has input time horizon $n > 16\sqrt{2} \frac{R_{\max}(R_{\max}^2 + \sigma^2)^{3/2} H_*^2 s^2}{\sigma^4} \log \frac{2d}{\delta}$, action set $\mathcal{A} \subset [-1, +1]^d$, and exploration length $n_0 = 4(s^2 \sigma^2 H_*^2 n^2 \log \frac{2d}{\delta} R_{\max}^{-2})^{\frac{1}{3}}$, $\lambda_1 = 4\sigma \sqrt{\frac{H_*^2}{n_0} \log \frac{2d}{\delta}}$, then with probability at least $1 - 2\delta$, $\mathrm{Reg}(n) \leq 8 R_{\max}^{1/3} (s^2 \sigma^2 H_*^2 n^2 \log \frac{2d}{\delta})^{\frac{1}{3}}$.

*Proof.* From Corollary 1, $\|\hat{\theta} - \theta^*\|_1 \leq 2s\lambda_1$ with probability at least $1 - 2\delta$. Therefore, with probability $1 - 2\delta$,

$$\mathrm{Reg}(n) \leq R_{\max} n_0 + (n - n_0)\|\hat{\theta} - \theta^*\|_1 \leq R_{\max} n_0 + 2sn\lambda_1 = R_{\max} n_0 + 8sn\sigma\sqrt{\frac{H_*^2}{n_0} \log \frac{2d}{\delta}}$$

and optimizing the right hand side with respect to $n_0$ leads to the desired upper bound. $\square$

Compared with Hao et al. [18]'s regret bound $\tilde{O}((R_{\max} s^2 \sigma^2 \mathcal{C}_{\min}^{-2} n^2)^{1/3})$[1], Algorithm 3's regret bound $\tilde{O}((R_{\max} s^2 \sigma^2 H_*^2 n^2)^{1/3})$ is at most $\tilde{O}((R_{\max} s^2 \sigma^2 \mathcal{C}_{\min}^{-1} n^2)^{1/3})$, which is at least a factor $\mathcal{C}_{\min}^{\frac{1}{3}}$ smaller. As we will see in Section 5, we show that the regret upper bound provided by Theorem 3 is unimprovable in general, answering an open question of [18].

**Improved upper bound with minimum signal condition.** Our second new algorithm, Algorithm 4, similarly uses WARM-POPART under an additional minimum signal condition.

**Assumption 2** (Minimum signal). There exists a known lower bound $m > 0$ such that $\min_{j \in \mathrm{supp}(\theta^*)} |\theta_j^*| > m$.

At a high level, Algorithm 4 uses the first $n_2$ rounds for identifying the support of $\theta^*$; the $\ell_\infty$ recovery guarantee of WARM-POPART makes it suitable for this task. Under the minimal signal condition and a large enough $n_2$, it is guaranteed that $\hat{\theta}_2$'s support equals exactly the support of $\theta^*$. After identifying the support of $\theta^*$, Algorithm 4 treats this as a $s$-dimensional linear bandit problem by discarding

---

[1]This is implicit in [18] – they assume that $\sigma = 1$ and do not keep track of the dependence on $\sigma$.

the remaining $d - s$ coordinates of the arm covariates, and perform phase elimination algorithm [23, Section 22.1] therein. The following theorem provides a regret upper bound of Algorithm 4.

---

**Algorithm 4** Restricted phase elimination with WARM-POPART

---

1: Input: time horizon $n$, finite action set $\mathcal{A}$, minimum signal $m$, failure rate $\delta$, reward threshold parameter $R_{\max}$, an upper bound of $\max_{a \in \mathcal{A}} |\langle \theta^*, a \rangle|$
2: Solve the optimization problem in Eq. 5 and denote the solutions as $Q$ and $\mu_*$, respectively.
3: Let $n_2 = \max\left(\frac{256\sigma^2 H_*^2}{m^2} \log \frac{2d}{\delta}, \frac{32s^2(R_{\max}^2 + \sigma^2)H_*^2}{\sigma^2} \log \frac{2d}{\delta}\right)$
4: **for** $t = 1, \ldots, n_2$ **do**
5:     Independently pull the arm $A_t$ according to $\mu_*$ and receives the reward $r_t$
6: **end for**
7: $\hat{\theta}_2 = $ WARM-POPART$(\{A_t\}_{t=1}^n, \{R_t\}_{t=1}^n, Q, \delta, R_{\max})$
8: Identify the support $\hat{S} = \mathrm{supp}(\hat{\theta}_2)$
9: **for** $t = n_2 + 1, \ldots, n$ **do**
10:     Invoke phased elimination algorithm for linear bandits on $\hat{S}$
11: **end for**

---

**Theorem 4.** If Algorithm 4 has input time horizon $n > \max\left(\frac{2^8 \sigma^2 H_*^2}{m^2}, \frac{2^5 s^2 (R_{\max}^2 + \sigma^2) H_*^2}{\sigma^2}\right) \log \frac{2d}{\delta}$, action set $\mathcal{A} \subset [-1, 1]^d$, upper bound of the reward $R_{\max}$, then with probability at least $1 - 2\delta$, the following regret upper bound of the Algorithm 4 holds: for universal constant $C > 0$,

$$\mathrm{Reg}(n) \le \max\left(\frac{2^8 \sigma^2 H_*^2}{m^2} \log \frac{2d}{\delta}, \frac{2^5 s^2 (R_{\max}^2 + \sigma^2) H_*^2}{\sigma^2} \log \frac{2d}{\delta}\right) + C\sigma\sqrt{sn\log(|\mathcal{A}|n)}$$

For sufficiently large $n$, the second term dominates, and we obtain an $O(\sqrt{sn})$ regret upper bound. Theorem 4 provides two major improvements compared to Hao et al. [18, Algorithm 2]. First, when $m$ is moderately small (so that the first subterm in the first term dominates), it shortens the length of the exploration phase $n_2$ by a factor of $s \cdot \frac{C_{\min}}{H_*^2}$. Second, compared with the regret bound $\tilde{O}\left(\sqrt{\frac{9\lambda_{\max}(\sum_{i=1}^{n_2} A_i A_i^\top / n_2)}{C_{\min}}}\sqrt{sn}\right)$ provided by [18], our main regret term $\tilde{O}(\sqrt{sn})$ is more interpretable and can be much lower.

## 5 Matching lower bound

We show the following theorem that establishes the optimality of Algorithm 3. This solves the open problem of Hao et al. [18, Remark 4.5] on the optimal order of regret in terms of sparsity and action set geometry in sparse linear bandits.

**Theorem 5.** For any algorithm, any $s, d, \kappa$ that satisfies $d \ge \max(n^{1/3} s^{4/3} \kappa^{-4/3}, (s+1)^2)$ and $n > 8\kappa s^2$, there exists a linear bandit environment an action set $\mathcal{A}$ and a $s$-sparse $\theta \in \mathbb{R}^d$, such that $C_{\min}^{-1} \le \kappa^{-2}$, $R_{\max} \le 2$, $\sigma = 1$, and

$$\mathrm{Reg}_n \ge \Omega\left(\kappa^{-2/3} s^{2/3} n^{2/3}\right).$$

We give an overview of our lower bound proof techniques, and defer the details to Appendix F.

**Change of measure technique.** Generally, researchers prove the lower bound by comparing two instances based on the information theory inequalities, such as Pinsker's inequality, or Bregtanolle-Huber inequality. In this proof, we also use two instances $\theta$ and $\theta'$, but we use the change of measure technique, to help lower bound the probability of events more freely. Specifically, for any event $A$,

$$\mathbb{P}_\theta(A) = \mathbb{E}_\theta[\mathbb{1}_A] = \mathbb{E}_{\theta'}\left[\mathbb{1}_A \prod_{t=1}^n \frac{p_\theta(r_t|a_t)}{p_{\theta'}(r_t|a_t)}\right] \gtrsim \mathbb{E}_{\theta'}\left[\mathbb{1}_A \exp\left(-\sum_{t=1}^n \langle A_t, \theta - \theta' \rangle^2\right)\right]. \tag{8}$$

**Symmetrization.** We utilize the algorithmic symmetrization technique of Simchowitz et al. [34], Stoltz et al. [37], which makes it suffice to focus on proving lower bounds against symmetric algorithms.

**Definition 2** (Symmetric Algorithm). An algorithm Alg is *symmetric* if for any permutation $\pi \in Sym(d)$, $\theta \in \mathbb{R}^d$, $\{a_t\}_{t=1}^n \in \mathcal{A}^n$,

$$\mathbb{P}_{\theta,\mathsf{Alg}}(A_1 = a_1, \cdots, A_n = a_n) = \mathbb{P}_{\pi(\theta),\mathsf{Alg}}(A_1 = \pi(a_1), \cdots, A_n = \pi(a_n))$$

where for vector $v$, $\pi(v) \in \mathbb{R}^d$ denotes its permuted version that moves $v_i$ to the $\pi(i)$-th position.

This approach can help us to exploit the symmetry of $\theta'$ to lower bound the right hand side of (8); below, $\Pi := \{\pi' : \pi(\theta') = \theta'\}$ is the set of permutations that keep $\theta'$ invariant, and $A$ is an event invariant under $\Pi$:

$$(8) \geq \frac{1}{|\Pi|}\sum_{\pi \in \Pi}\mathbb{E}_{\theta'}\left[\mathbb{1}_A\exp(-\sum_{t=1}^n\langle\pi^{-1}(A_t),\theta-\theta'\rangle^2)\right] \geq \mathbb{E}_{\theta'}\left[\mathbb{1}_A\exp\left(-\sum_{t=1}^n\frac{1}{|\Pi|}\sum_{\pi\in\Pi}\langle\pi^{-1}(A_t),\theta-\theta'\rangle^2\right)\right]$$

which helps us use combinatorial tools over the actions for the lower bound proof.

## 6 Experimental results

We evaluate the empirical performance of POPART and our proposed experimental design, along with its impact on sparse linear bandits. One can check our code from here: `https://github.com/jajajang/sparse`.

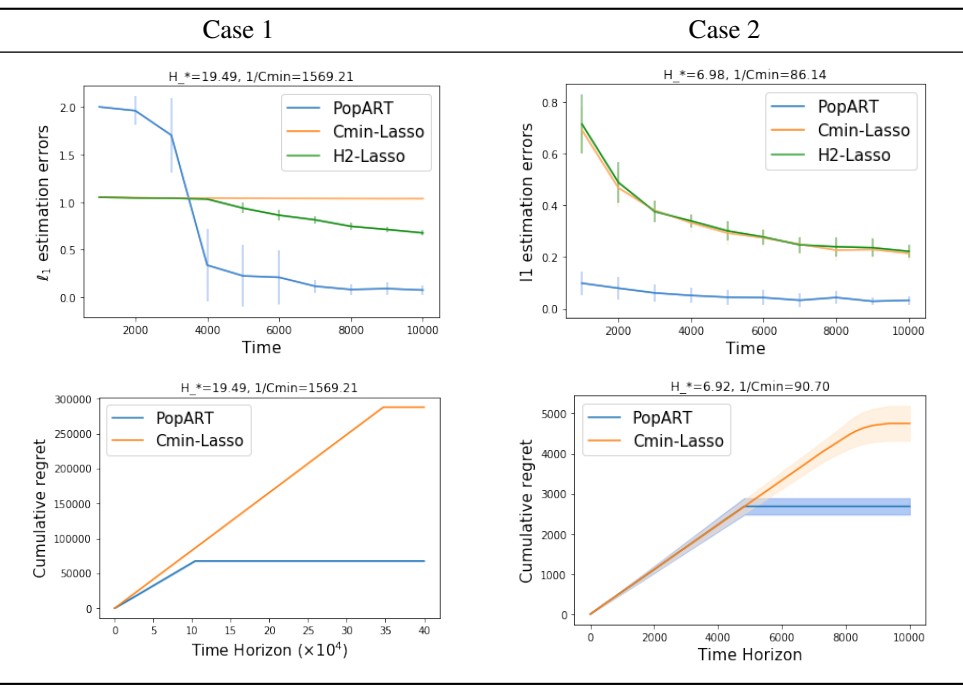

Figure 1: Experiment results on $\ell_1$ estimation error cumulative regret.

For sparse linear regression and experimental design, we compare our algorithm POPART with $\mu$ being the solution of (5) with two baselines. The first baseline denoted by $C_{\min}$-Lasso is the method proposed by Hao et al. [18] that uses Lasso with sampling distribution $\mu$ defined by (6). The second baseline is $H^2$-Lasso, uses Lasso with sampling distribution $\mu$ defined by (5), which is meant to observe if Lasso can perform better with our experimental design and to see how POPART is compared with Lasso as an estimator since they are given the same data. Of course, this experimental design is favored towards POPART as we have optimized the design for it, so our intention is to observe if there ever exists a case where POPART works better than Lasso.

For sparse linear bandits, we run a variant of our Algorithm 3 that uses WARM-POPART in place of POPART for simplicity. As a baseline, we use ESTC [18]. For both methods, we use the exploration length prescribed by theory. We consider two cases:

- **Case 1: Hard instance where $H_*^2 \ll \mathcal{C}_{\min}^{-1}$.** We use the action set constructed in Appendix D.1 where $H_*^2$ and $\mathcal{C}_{\min}$ shows a gap of $\Theta(d)$. We choose $d = 10$, $s = 2$, $\sigma = 0.1$.
- **Case 2. General unit vectors.** In this case, we choose $d = 30$, $s = 2$, $\sigma = 0.1$ and the action set $\mathcal{A}$ consists of $|\mathcal{A}| = 3d = 90$ uniformly random vectors on the unit sphere.

We run each method 30 times and report the average and standard deviation of the $\ell_1$ estimation error and the cumulative regret in Figure 1.

**Observation.** As we expected from the theoretical analysis, our estimator and bandit algorithm outperform the baselines. In terms of the $\ell_1$ error, for both cases, we see that POPART converges much faster than $\mathcal{C}_{\min}$-Lasso for large enough $n$. Interestingly, $H^2$-Lasso also improves by just using the design computed for POPART in case 1. At the same time, $H^2$-Lasso is inferior than POPART even if they are given the same data points. While the design was optimized for POPART and POPART has the benefit of using the population covariance, which is unfair, it is still interesting to observe a significant gap between POPART and Lasso. For sparse linear bandit experiments, while ESTC requires exploration time almost the total length of the time horizon, ours requires a significantly shorter exploration phase in both cases and thus suffers much lower regret.

## 7  Conclusion

We have proposed a novel estimator POPART and experimental design for high-dimensional linear regression. POPART has not only enabled accurate estimation with computational efficiency but also led to improved sparse linear bandit algorithms. Furthermore, we have closed the gap between the lower and upper regret bound on an important family of instances in the data-poor regime.

Our work opens up numerous future directions. For POPART, we speculate that $(Q(\mu)^{-1})_{ii}$ is the statistical limit for testing whether $\theta_i^* = 0$ or not – it would be a valuable investigation to prove or disprove this. We believe this will also help investigate whether the dependence on $H_*^2$ in our regret upper bound is unimprovable (note our matching lower bound is only for a particular family of instances). Furthermore, it would be interesting to investigate whether we can use POPART without relying on the population covariance; e.g., use estimated covariance from an extra set of unlabeled data or find ways to use the empirical covariance directly. For sparse linear bandits, it would be interesting to develop an algorithm that achieves the data-poor regime optimal regret and data-rich regime optimal regret $\sqrt{sdn}$ simultaneously. Furthermore, it would be interesting to extend our result to changing arm set, which poses a great challenge in planning.

## Acknowledgments and Disclosure of Funding

We thank Ning Hao for helpful discussions on theoretical guarantees of Lasso. Kwang-Sung Jun is supported by Data Science Academy and Research Innovation & Impact at University of Arizona.

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
