# OpenReview forum: "PopArt: Efficient Sparse Regression and Experimental Design for Optimal Sparse Linear Bandits"
_NeurIPS.cc/2022/Conference — NeurIPS 2022 Accept_

### Official Review · Reviewer_7Qj2 · 2022-07-10

**Rating:** 6
**Confidence:** 4
**Soundness:** 3 good
**Presentation:** 4 excellent
**Contribution:** 4 excellent

**Summary:**

This paper revisited the analysis of the sparse regression and the sparse linear bandit problem.
Firstly, they presented the improved linear regression algorithm POPART by combining the idea of Catoni's estimator and the thresholding.
Then, they showed that using POPART, a simple explore then commit type algorithm improves the regret bound of Hao et al in the data-poor regime.
Thirdly, they showed that by using the POPART for the support estimate and running a linear bandit algorithm, they presented an improved regret upper bound for the data-rich regime.
Moreover, a tighter lower bound for the data-poor regime is presented.


**Questions:**



### general questions

(Although the same is true of Hao et al.) the action set has to span the entire high dimension so the requirement for the action set can be a bit strict for the high dimensional setting.
The compatibility constant is, in this sense, a much weaker assumption as it does not require the lower bound on the minimum eigenvalue.
In the results, the lower bound condition in Hao et al is removed but the terms with the diagonal components of Q are used. Does this mean that the lower bound on the eigenvalue condition can be eliminated? Or do you still need to assume that the action set has to span R^d? My impression was the algorithm utilized the idea of Hao et al, but thanks to the improvement with POPART, the requirement on the action set can be further relaxed.


 ### minor, typo
line 53, which can is

I think subgaussian random variable should have zero-mean and no need to state "zero-mean \sigma-subgaussian".


Eq. (3) I think \lambda is not defined so far.

Algorithm 2, Samples are indexed until n, but should be n_0.

Algorithms 1-4,  \sigma is also an input.

Algorithm 4, the phased elimination algorithm is not defined in the main paper. Should cite reference or write pointers.

Eq. (6) I think \lambda_min is not defined.

line 228 and and






**Limitations:**

See Questions.

**Strengths And Weaknesses:**

The main contribution of this paper is in its theoretical aspect.
Novel algorithm/analysis of the linear regression and it's consequence is significant.
Their guarantee has tighter dependence than the classical Lasso guarantee by focusing on the different quantities (especially Q^{-1}_{ii}).

With the help of the novel offline regression analysis, the regret bounds for the sparse linear bandit literature are further improved. For the bandit upper bound part, the algorithmic part may not be so novel but the resulting tighter analysis is significant.
They further presented a tighter lower bound for the data-poor regime with the novel application of the change of measure technique with the symmetrization approach, this technique is noteworthy.

Although there may be some parts of the paper that can be improved, I think the paper is worth accepted for Neurips.

---

> ### Author Response · Authors · 2022-08-02
> **Response to the reviewer**
>
>
> Thank you for your feedback and questions. We will incorporate your comments on typos in the final version.
>
> >In the results, the lower bound condition in Hao et al is removed but the terms with the diagonal components of Q are used. Does this mean that the lower bound on the eigenvalue condition can be eliminated? Or do you still need to assume that the action set has to span $R^d$?
>
> For your main quesiton on the action set that does not span the whole $\mathbb{R}^d$:
>
> * For our current algorithm, we need to take the inverse of $Q(\mu)$. Therefore, we need the condition that the action set spans $\mathbb{R}^d$ since $Q(\mu)$ should be invertible.
>
>
> * Some other sparse linear bandit algorithms can still deal with arm sets that do not span $\mathbb{R}^d$ if one is fine with regret order of $\sqrt{sdn}$, which is meaningful in the data-rich regime only. See the 'OFUL with SeqSEW' algorithm of 'Online-to-Confidence-Set Conversions and Application to Sparse Stochastic Bandits' by Abbasi-Yadkori et al. (2012) for an example.
>
>
> >(Although the same is true of Hao et al.) the action set has to span the entire high dimension so the requirement for the action set can be a bit strict for the high dimensional setting. The compatibility constant is, in this sense, a much weaker assumption as it does not require the lower bound on the minimum eigenvalue.
>
> For comparison between PopArt and Lasso in the sparse linear bandit settings:
> * In the estimation perspective where the data is given and one has to calculate the optimal estimator, we agree that it is possible that Lasso-based linear bandit algorithms may be applicable to linearly degenerate action spaces, whereas PopArt cannot.
> * However, in the sparse linear bandit and experimental design perspective, experimental design using the compatibility condition criterion (Eq. (2) in our paper) suffers from a computational issue: the compatibility constant is computationally intractable to calculate, at least naively, let alone designing a distribution $\mu$ that maximizes it. In this sparse linear bandit setting, this intractability of calculating the compatibility constant also makes the learner hard to calculate an optimal exploration schedule. See also the discussions in lines 186-190 in our paper.

---

> > ### Comment · Reviewer_7Qj2 · 2022-08-09
> > **Re: Response to the reviewer**
> >
> > Thank you very much for your detailed reply.
> >
> > I have one question regarding your reply.  Why do you mention the complexity of computing the compatibility constant? As far as I know, no algorithm requires the computation of the compatibility constant in their algorithms. I believe compatibility constant is only required for the theoretical guarantees.
> > Having an assumption on the compatibility constant does not mean that an optimal algorithm should compute the compatibility constant.
> >
> > Best wishes,
> >
> > Reviewer 7Qj2

---

> > > ### Author Response · Authors · 2022-08-09
> > > **Re: Re: Response to the reviewer**
> > >
> > > Thank you for following up!
> > > I guess saying 'comparison between PopArt and Lasso' was a bit confusing. It's more like comparing how one can do design of experiments with PopArt vs Lasso.
> > >
> > > To answer your question, it depends on what 'algorithm' you mean. Two possibilities are: (a) an algorithm that computes an estimator given a data generated from an external source and (b) an algorithm that designs a distribution over the arm set, sample from it and observe their labels, and then computes and estimator.
> > >
> > > We meant (b) in our response. We aim to have an experimental design algorithm for minimizing the L1 recovery error. Specifically, the L1 recovery error of Lasso depends on the compatibility constant, so it is natural to find a design (i.e., probability distribution over the arms) that would minimize the compatibility constant. Of course, you can just use an arbitrary design like the uniform probability distribution, but there is no guarantee that it will work well. Hao et al. (2020) go around the computational difficulty issue by maximizing the minimum eigenvalue of the design, but it results in an inferior bound than our proposed method.
> > >
> > > If you meant (a), then, yes, you do not need to compute the compatibility constant.

---

### Official Review · Reviewer_Vvdf · 2022-07-12

**Rating:** 7
**Confidence:** 3
**Soundness:** 4 excellent
**Presentation:** 3 good
**Contribution:** 3 good

**Summary:**

The paper studies multi-armed linear bandits in high dimensions under sparsity
assumptions. This problem has been studied before (notably by Hao et al '20), but the
paper presents newer (tighter) upper and lower bounds. Key to the algorithm is a novel
estimator for sparse linear regression which allows us to design a sampling distribution
so as to minimize the estimation error. While a s^(2/3) T^(2/3) regret (where s is the
sparsity and T is the time horizon) seems unavoidable in general, the authors show that
\sqrt{sT} rate is possible with a modified version of the algorithm if there is enough
signal from the non-zero coefficients. Finally the authors present some simple
simulations to corroborate their experimental results.

While the results are not earth-shattering, the paper makes solid progress on a
well-established and useful problem. While I didn't have the time to read the proofs in
detail (hence the lower confidence score), the results appear believable and the
intuitions are presented clearly. Hence, I would like to see this paper accepted.


**Questions:**

See above.

**Limitations:**

See above.

**Strengths And Weaknesses:**

I don't have any major comments/criticisms about the paper. But here are some suggestions
which might help improve the presentation.
1. There were too many in-line equations, and generally, the presentation was somewhat
equation-driven. While I appreciate that this may be unavoidable for a paper such as this,
I felt that the authors could have done a better job of making the material easier to
read.
2. Please try and use words to describe symbols/notation when they are being introduced
(e.g: Explain what R0 is in assumption 1 ).
3. Maybe introduce the compatibility condition (equation 2) closer to the discussion
around Theorem 2. Right now, it doesn't directly relate to the results in the paper and I
was left confused as to what it's purpose was.
4. Instead of using large equations in the algorithm (e.g. line 6), consider defining the
quantities in the text and referring to them in the algorithm pseudo-code.
5. Please explain why knowledge of the signal strenght is necessary for an algorithm to
achieve \sqrt{T} regret around Theorem 4.

---

> ### Author Response · Authors · 2022-08-02
> **Response to the reviewer**
>
> Thank you for your comments on improving the readability of the paper. We will incorporate your comments in the final version.
>
> >Please explain why knowledge of the signal strenght is necessary for an algorithm to achieve $\sqrt{T}$ regret around Theorem 4.
>
> On item 5, For the algorithmic perspective, we need the minimum signal condition to ensure the length of exploration is long enough so that we can exactly recover $\mathrm{supp}(\theta^*)$. This fact helps us to decrease the length of exploration compared with Algorithm 3 and hence achieve a $\sqrt{n}$ regret bound.
>
> If you are asking about whether it is possible to achieve $\sqrt{n}$ regret bound using Algorithm 4 without the knowledge of $m$, the answer is yes: we can do so by modifying the exploration length of Algorithm 4 to, say, $n_2 = \sqrt{n}$. As long as $n$ is larger than a constant ( $n> \frac{2^{16} H_\star^{4} \sigma^4 }{m^4} \log^4 \frac{2d}{\delta}$ ), we still guarantee that $\hat{S} = \mathrm{supp}(\theta^*)$ with high probability, and we can still enjoy the same order of $O(s\sqrt{n})$ regret.

---

### Official Review · Reviewer_ToPD · 2022-07-12

**Rating:** 7
**Confidence:** 3
**Soundness:** 3 good
**Presentation:** 3 good
**Contribution:** 3 good

**Summary:**

The authors consider the sparse linear regression problem with an application to experimental design in linear bandits. They propose a new sparse linear estimator of the true model based on Catoni's estimator assuming the knowledge of the population covariance.
The authors bound the error of the proposed estimator in terms of the spectral properties of the covariance matrix. They also propose a warm start estimator and improve the error bound further. Using the error bound of this estimator, the authors then propose a criterion for designing sampling distribution in the linear bandits problem. Using this sampling distribution, they propose an explore-then-commit algorithm for experimental design in linear bandits. The authors then bound the regret of this strategy and further provides a matching lower bound. Finally, under an additional assumption on the minimum signal of the unknown model, they design a phase-based algorithm using the estimator and provide its regret bound.

**Questions:**

It would be great if a discussion on implementation of the convex optimization problem is presented.

**Limitations:**

Yes

**Strengths And Weaknesses:**

Strengths:
1. The authors provide a novel estimator for sparse linear regression and prove a tighter error bound for it compared to well-known Lasso estimator.

2. The tighter error bound for the estimator also leads to a tighter regret bound for linear bandits.

3. They also improve the lower bound for experimental design in linear bandits compared to prior work and establish that it is near to minimax optimal.

4. The results are well-positioned compared to the existing work and the paper is well-written.

Weakness:

The bandit algorithm for experimental design needs to solve an optimization problem to find the sampling distribution. I find the implementation details and complexity of that procedure missing from the paper.

---

> ### Author Response · Authors · 2022-08-02
> **Response to the reviewer**
>
>
> Thanks for your positive review and valuable feedback!
>
> First, to prove that the optimization problem that defines $H_*^2$ (Eq. (5)) is an convex optimization, we show that the mapping $f(\\mu) = \\max_i ((Q(\mu))^{-1})_{i,i}$
>
> is convex. It suffices to show that $g(X) = \max_i (X^{-1})_{i,i}$ is convex.
>
>
> To see this, first note that the inverse of a matrix is convex, i.e. for two positive definite matrices $X, Y$,
>
> $$ ((1-\lambda)X + \lambda Y )^{-1} \preceq (1-\lambda) X^{-1} + \lambda Y^{-1}$$
>
> and therefore maximum entry of the inverse is also a convex function since
> \begin{align*} \max_{i \in [d]} e_i^\top ((1-\lambda)X + \lambda Y )^{-1} e_i & \leq \max_{i \in [d]} e_i^\top [(1-\lambda) X^{-1} + \lambda Y^{-1}] e_i \\\\ &\leq (1-\lambda) \max_{i \in [d]} e_i^\top X^{-1} e_i + \lambda \max_{i \in [d]} e_i^\top Y^{-1} e_i
> \end{align*}
>
> Therefore, we can apply traditional convex optimization methods to solve the optimization problem that defines $H_*^2$ (Eq. (5)).
>
> To implement this, we solve the following optimization problem.
>
> \begin{align*}
> \min_{\mu \in \Delta^d , T \in \mathcal{S}^{d\times d}} &\max_{i \in [d]} T_{ii}\\\\
> \text{Subject to } & T \succeq Q(\mu)^{-1}
> \end{align*}
>
> where $\Delta^d:=\\{ v\in \mathbb{R}_+^d: \|v\|_1 = 1\\}$ and $\mathcal{S}^{d \times d}$ is the set of symmetric matrices. We used CVXPY, a popular convex optimization tool in Python, with code as follows:
> ```
> import cvxpy as cp
> import numpy as np
> %d is dimension
> %A is our action set matrix
> mu=cp.Variable(d, pos=True)
> Q=A.T@ cp.diag(mu) @A
> T = Variable((d, d), symmetric=True)
> M = bmat([[Q, np.eye(d)],
>           [np.eye(d), T]])
> constraints = [M >> 0, cp.sum(mu)==1]
> objective = cp.Minimize(cp.max(cp.diag(T)))
> prob=cp.Problem(objective, constraints)
> prob.solve(solver=cp.MOSEK,verbose=True)
> ```
>
> Here, we used the fact that $M\succeq 0 \Leftrightarrow T \succeq Q^{-1}$. Below is a proof of this fact:
>
> First, note that:
> $M \succeq 0 \Leftrightarrow v^\top M v \geq 0, \forall v \in \mathbb{R}^{2d} \Leftrightarrow u^\top Q u + v_{d+1:2d}^\top (T-Q^{-1}) v_{d+1:2d}\geq 0$
> where $u=v_{1:d}+Q^{-1}v_{d+1:2d}, \forall v \in \mathbb{R}^{2d}$.
>
> Now:
> * $(\Leftarrow)$ If $T-Q^{-1}\succeq 0$, then by the above observation, $M \succeq 0$ obviously holds.
> * $(\Rightarrow)$ If $T-Q^{-1}$ is not positive semidefinite, then there exists $v'\in \mathbb{R}^{d}$ such that $(v')^\top (T-Q^{-1})v'< 0$. Let $v_f = -Q^{-1}v'$ and $v=\begin{bmatrix}v_f  \\ v'\end{bmatrix}$ then $v^\top M v <0$, showing that $M$ is also not positive semidefinite.
>
>
>
> About the computational complexity, CVXPY uses the MOSEK solver, which in turn uses primal-dual interior point method, which has a polynomial running time.
>
> We will add these discussions in the final version.

---

### Meta-Review · Area_Chair_LvBd · 2022-08-23

**Recommendation:** Accept
**Confidence:** Certain

**Metareview:**

The paper is motivated by the design of low-regret algorithms for high-dimensional sparse linear bandit problems. The challenge is to obtain regret guarantees even in the data-poor regime where the number of samples the learner can gather may be smaller than the dimension.

This challenge had been investigated in [12] with a regret scaling as $(sn/C_{min})^{2/3}$ ($s$ is the sparsity of the problem, $n$ the number of samples, $C_{min}$ is the maximum over all possible arm distribution of the resulting average variance. The authors propose a scheme whose regret scales at most as  $(sn H)^{2/3}$ where $H$ is a new (minimax) constant, proven to be smaller than $1/C_{min})$. The paper also presents a matching minimax regret lower bound.

To achieve this improved regret upper bound, the authors develop a new parameter estimation procedure, based notably on Catoni’s estimator (this kind of estimator has been recently advocated in RL with linear function approximation, see “Reward-Free RL is No Harder Than Reward-Aware RL in Linear Markov Decision Processes”, Wagenmaker et al., ICML 2022, and the authors could mention this paper and stress the differences in the use of this estimator). The derivation of the lower bound also relies on new techniques (as mentioned by one of the reviewers).
Overall, this is a solid contribution, even though compared to [12], the improvement is not that spectacular.


**Award:**

No

---

### Decision · Program_Chairs · 2022-09-14

Accept